# Intercorrelation of Molecular Biomarkers and Clinical Phenotype Measures in Fragile X Syndrome

**DOI:** 10.3390/cells12141920

**Published:** 2023-07-24

**Authors:** Ramkumar Aishworiya, Mei-Hung Chi, Marwa Zafarullah, Guadalupe Mendoza, Matthew Dominic Ponzini, Kyoungmi Kim, Hazel Maridith Barlahan Biag, Angela John Thurman, Leonard Abbeduto, David Hessl, Jamie Leah Randol, Francois V. Bolduc, Sebastien Jacquemont, Sarah Lippé, Paul Hagerman, Randi Hagerman, Andrea Schneider, Flora Tassone

**Affiliations:** 1MIND Institute, University of California Davis Medical Center, Sacramento, CA 95817, USA; paearam@nus.edu.sg (R.A.); mdponzini@ucdavis.edu (M.D.P.); hbbiag@ucdavis.edu (H.M.B.B.);; 2Khoo Teck Puat-National University Children’s Medical Institute, National University Health System, Singapore 119074, Singapore; 3Department of Pediatrics, Yong Loo Lin School of Medicine, National University of Singapore, Singapore 119228, Singapore; 4Department of Psychiatry, National Cheng Kung University Hospital, Tainan 704, Taiwan; 5Department of Biochemistry and Molecular Medicine, School of Medicine, University of California Davis, Sacramento, CA 95817, USAgumendoza@ucdavis.edu (G.M.);; 6Department of Public Health Sciences, School of Medicine, University of California Davis, Sacramento, CA 95817, USA; 7Department of Pediatrics, School of Medicine, University of California Davis, Sacramento, CA 95817, USA; 8Department of Psychiatry and Behavioral Sciences, School of Medicine, University of California Davis, Sacramento, CA 95817, USA; 9Integrative Genetics and Genomics Graduate Group, University of California Davis, One Shields Avenue, Davis, CA 95616, USA; 10UC Davis Biotechnology Program, University of California Davis, Davis, CA 95616, USA; 11Department of Pediatrics, Department of Medical Genetics, Women and Children Health Research Institute, University of Alberta, Edmonton, AB T6G 2R3, Canada; 12CHU Sainte-Justine Research Center, Université de Montréal, Montreal, QC H3T 1J4, Canada; 13Department of Pediatrics, University of Montreal, Montreal, QC H3T 1J4, Canada; 14Department of Psychology, Université de Montréal, Montreal, QC H3T 1J4, Canada

**Keywords:** fragile X syndrome, *FMR1* mRNA, MMP9, FMRP, clinical trial, outcome measures

## Abstract

This study contributes to a greater understanding of the utility of molecular biomarkers to identify clinical phenotypes of fragile X syndrome (FXS). Correlations of baseline clinical trial data (molecular measures—*FMR1* mRNA, *CYFIP1* mRNA, MMP9 and FMRP protein expression levels, nonverbal IQ, body mass index and weight, language level, NIH Toolbox, adaptive behavior rating, autism, and other mental health correlates) of 59 participants with FXS ages of 6–32 years are reported. *FMR1* mRNA expression levels correlated positively with adaptive functioning levels, expressive language, and specific NIH Toolbox measures. The findings of a positive correlation of MMP-9 levels with obesity, *CYFIP1* mRNA with mood and autistic symptoms, and *FMR1* mRNA expression level with better cognitive, language, and adaptive functions indicate potential biomarkers for specific FXS phenotypes. These may be potential markers for future clinical trials for targeted treatments of FXS.

## 1. Introduction

Fragile X syndrome (FXS), an X-linked dominant neurodevelopmental disorder, is the most prevalent inherited cause of intellectual disability (ID) and the leading single-gene cause of autism spectrum disorder (ASD) [1,2]. The population prevalence of FXS in the United States is estimated to vary from 1 in 5000 to 7000 males and from 1 in 8000 to 11,000 females [3,4]. Individuals with FXS have characteristic physical features, including long facies, prominent ears, and macroorchidism, as well as medical and neuropsychiatric comorbidities, such as connective tissue disorders, sleep disturbances, seizures, inattention, hyperactivity, anxiety, aggression, self-injury, and stereotypical behaviors [5,6,7]. The neurocognitive profile of individuals with FXS often includes language impairment in addition to executive dysfunction with impaired inhibition, working memory, cognitive flexibility, and planning, although there is considerable within-syndrome heterogeneity [8,9]. These impairments have been shown to influence social behaviors, mood, and adaptive skills as well [10,11].

The pathophysiology of FXS is attributed to an expansion in the number of CGG repeats in the 5′untranslated region of the fragile X messenger ribonucleoprotein 1 (*FMR1*) gene [5]. An expansion greater than 200 repeats causes aberrant DNA methylation on the *FMR1* promotor, which then leads to gene silencing and the absence or loss of function of its encoded protein, the *FMR1* protein (FMRP). FMRP is a crucial protein with multiple functions, including regulation of mRNA translation, modulation of multiple ionic channels, and regulation of neuronal synaptic function, memory, and plasticity [12,13]. The understanding of the pathways disrupted and the nature of the proteins influenced by FMRP, and their implications on the clinical phenotypes and other neuropsychological functions is important since current interventions for FXS are mainly supportive, with curative ones under development. Exploration of potential molecular biomarkers is necessary for the development of targeted treatment protocols for FXS [14,15,16].

Identification of suitable molecular biomarkers that reliably correlate with the clinical phenotype is an ongoing challenge in the field of FXS research [17]. Among these candidate biomarkers, *FMR1* mRNA and FMRP expression levels and percentage of methylation have been studied extensively and have been shown to associate with cognitive and executive function [15,18,19,20]. The number of CGG repeats and the X-activation ratio have also been related to intelligence, executive function, visual–spatial perception, and physical features [21]. However, there are considerable challenges with using *FMR1* mRNA and FMRP expression levels as blood levels of these measures may not accurately represent intra-cellular neuronal values, more directly related to brain function. Further, data on the implications of FMRP and *FMR1* mRNA levels on the behavioral profile of FXS are also limited. Hence, downstream targets of FMRP, including excitatory glutamatergic and inhibitory GABAergic pathways and their associated proteins, have been studied as targets for treatments in FXS [22,23]. This has included proteins involved in cellular signaling, such as phosphoinositide 3-kinase (PI3K), extracellular-regulated kinase (ERK), mammalian target of rapamycin (mTOR), matrix metalloproteinase-9 (MMP-9), brain-derived neurotrophic factor (BDNF), and amyloid-β protein precursor (APP), which have been the targets of many preclinical studies [24,25,26,27,28]. Among these, MMP-9 has been a promising target for intervention, given that it is secreted in the extracellular space, making blood plasma levels more reliable. Higher plasma MMP-9 levels have been reported in individuals with FXS compared to controls [27]. FMRP regulates the translation of MMP-9 mRNA at the synapses [29]. Although MMP-9 has been previously associated with the presence of seizures, learning disorders, and anxiety in mouse models, its implication on cognition, behavior, and functioning is currently not well-defined [30,31]. The cytoplasmic FMRP interacting protein (*CYFIP1*) interacts with FMRP to form an inhibitory complex that regulates long-term synaptic plasticity. Interestingly, the *CYFIP1* gene has been identified as a candidate risk gene for autism [32,33], and significantly decreased levels of *CYFIP1* mRNA have been observed in a subgroup of individuals with FXS and the Prader–Willi-like phenotype, as compared to controls, although the clinical phenotypic implications of this observation are not yet fully clear [34].

The aim of the present study was to compare the associations among clinical measures and to examine the relationship with specific molecular measures (*FMR1* mRNA, FMRP, MMP-9, *CYFIP1* mRNA) in individuals with FXS and their clinical phenotype in terms of cognition, adaptive skills, language, behavioral profile and quality of life. These molecular measures were chosen based on their potential as biomarkers in FXS, and the range of clinical measures was intentionally comprehensive to identify the potentially varied relationships between the molecular and clinical measures. We have also assessed the intercorrelations among the clinical measures, including those recently proposed for use as outcome measures in clinical trials involving individuals with FXS [35,36], to further ascertain their utility for inclusion in clinical trials for this population. 

## 2. Materials and Methods

### 2.1. Study Sample and Procedure

Data for this study were obtained as part of an ongoing clinical trial involving individuals with FXS at a tertiary academic institution (clinicaltrials.gov identifier: NCT03479476). Pertinent inclusion criteria included the following: 1. diagnosis of FXS confirmed by previous genetic testing with a full mutation at the *FMR1* gene (≥200 CGG repeats); 2. chronological age between 6 and 40 years inclusive; 3. nonverbal IQ as assessed by the Leiter-3 (see below) of <85; and 4. able to speak at least occasional 3-word phrases by caregiver report. The presence of co-occurring conditions common to FXS, including attention-deficit hyperactivity disorder, anxiety, and depression, were not exclusion criteria; however, individuals with any serious chronic systemic medical illness were excluded. Ethics approval for the trial was obtained from the institutional review board, and depending on their age and cognitive capacity, all individuals with FXS or their parents provided written informed consent for participation. Study participants and/or their parents/caregivers were administered the various study measures over 2 days as part of assessments to determine eligibility for the clinical trial. Data from these baseline assessments were used in this manuscript. 

### 2.2. Study Measures

The following standardized measures were administered to all study participants at the beginning of the study: 

Vineland Adaptive Behavior Scales—Third Edition (VABS-3)—The VABS-3 [37] is a widely used assessment of everyday functioning that provides a standard score in the communication, daily living, social skill domains, and an adaptive behavior composite. The VABS-3 has been well validated with good internal consistency, including for use in individuals with intellectual disability. The VABS-3 was administered in an interview format by research personnel, and the informant was a parent/caregiver. Higher scores in each domain and in the adaptive behavior composite reflect higher adaptive skills; 

The Leiter-3—The Leiter-3 [38] is a standardized measure of non-verbal cognition that can be used with individuals with limited expressive language; it provides a standard score that has a norm of 100 and a standard deviation of 15. It allowed for an accurate measure of cognition among participants who spoke various languages, including non-English language. The non-verbal IQ composite score was used for analysis, with higher scores indicating higher cognitive skills. The average IQ range falls between 85–115;

Expressive Language Sampling (ELS)—The ELS is a validated measure for assessing a range of expressive language skills in individuals with FXS and intellectual disability [35]. ELS is administered by a trained examiner who interacts with the participant to elicit brief samples of conversational language (on a standard list of topics including school, vacations, and games) and narrative language (using a wordless picture book). Using a script for prompts and his/her responses, the examiner minimizes their participation, maximizes the participant’s contribution, and avoids the use of examiner language that would unduly constrain the participant’s talk. Audio-recorded language samples are transcribed and analyzed using specialized software to generate scores for Narration, Conversation, and a Composite of both sampling contexts. In the present study, we computed a measure of the diversity of vocabulary used (i.e., the number of different word roots used up to a maximum of 50 C-units, with a C-unit defined as an utterance ranging from a single word up to and including an independent clause and its modifiers). ELS has been used previously in several studies of individuals with FXS and shown to have high test–retest reliability, minimal practice effects, and strong construct validity for the vocabulary measure, including in those with low cognitive abilities [39];

Autism Diagnostic Observation Schedule (ADOS-2)—The ADOS-2 [40] is a semi-structured standardized assessment that is one of the gold standard measures for the diagnosis of autism spectrum disorder (ASD). The ADOS-2 is a play-based assessment that examines features of ASD; it includes play-based tasks and/or questions designed to elicit social communication skills and repetitive behaviors, if any. In this study, this was administered by trained, research-reliable study personnel. Modules of the ADOS-2 were selected as appropriate based on the participant’s age and language levels. A total score, as well as domain scores for social communication and behavior impairment, were generated. Based on these scores, the presence of ASD was determined and further classified into severity categories of low, moderate, and high severity;

The NIH Toolbox Cognition Battery (NIHTB-CB)—The NIH Toolbox Cognition Battery [41] is psychometrically sound, iPad-administered, and developed for research under the consensus of several NIH Institutes. It is adaptative, efficient, and can be used across the lifespan. The NIHTB-CB includes seven measures with cognitive constructs of attention, executive function, episodic memory, working memory, language, and processing speed. The seven measures are the Dimensional Change Card Sort Test (for executive function-cognitive flexibility), the Flanker Inhibitory Control and Attention Test (for executive function-inhibitory control and sustained attention), the Picture Sequence Memory Test (for episodic memory and visual memory), the Picture Vocabulary Test (for language–vocabulary comprehension), the Oral Reading Recognition Test (for language–reading decoding), the List Sorting Working Memory Test (for working memory), and Pattern Comparison Processing Speed Test (for processing speed). The NIHTB-CB has been extensively evaluated in children and young adults with Intellectual Developmental Disabilities (IDD), including FXS [36,42,43]. These tests are performance-based. Higher scores reflect better performance in each task;

The Aberrant Behavior Checklist—Community Edition (ABC-C)—The ABC-C is a caregiver-completed 58-item questionnaire that measures challenging behaviors in populations with intellectual and developmental disabilities. It has been studied in individuals with FXS as well, with a validated FXS-specific factoring system [44]. Results are in the form of 6 subscales, namely, irritability, lethargy, social avoidance, stereotypic behavior, hyperactivity, and inappropriate speech. Higher scores represent greater challenging behaviors;

The Anxiety Depression and Mood Screen (ADAMS)—The ADAMS [45] is a parent/caregiver report, 28-item questionnaire, that screens for the presence of symptoms of anxiety, mood, and depression among individuals with intellectual disability. The ADAMS yields 5 subscale scores: General Anxiety; Social Avoidance; Depression; Manic/Hyperactive and Obsessive/Compulsive Behavior. Higher scores represent the presence of more symptoms of each sub-scale;

Swanson, Nolan, and Pelham Questionnaire (SNAP-IV)—The SNAP-IV [46] is a standardized caregiver-completed questionnaire based on the DSM-5 criteria that measure the symptoms of attention deficit hyperactivity disorder. Results include sub-scale scores for inattention, hyperactivity/impulsivity, and combined symptoms. The higher the score, the higher the degree of hyperactivity and inattention; 

Pediatric Quality of Life Questionnaire (PedsQL) Parent Proxy—The PedsQL [47] is a well-validated caregiver-reported measure of the quality of life of children and adolescents. It provides an overall quality of life score and sub-scale scores in physical functioning, emotional functioning, social functioning, and school functioning, with higher scores indicating better quality. The parent proxy module designed for children of 8–12 years of age was administered to the caregivers of all subjects, regardless of age, because the questions in this version were most appropriate for the overall study population’s cognitive age and ability. For any subjects not in school, questions pertaining to “school” were replaced with references to “work” or other activities in their life. Higher scores indicate better-reported quality of life;

Child Sleep Habits Questionnaire (CSHQ)—The CHSQ [48] is a standardized measure of sleep problems and consists of a series of 50 questions relating to sleep. This was completed by the parent/caregiver of the participant. The CHSQ provides a total sleep score and further scores under the domains of sleeping arrangement, sleeping duration, sleep routine, sleep resistance, night awakenings, parasomnias, sleep-disordered breathing, morning behavior, daytime behavior, and parental perception. Higher scores indicate greater levels of sleep-related problems in each domain.

Molecular measures

CGG sizing: Genomic DNA was isolated from whole blood using standard procedures (Qiagen, Valencia, CA, USA). CGG allele sizing was achieved by PCR and Southern Blot analysis, as previously reported [49,50]. Densitometric analysis was used to determine the percent of methylation, including the percentage of methylated alleles, and in females, the activation ratio (AR), which expresses the percentage of cells carrying the normal allele on the active X chromosome and measured as described in Tassone et al., 1999 [51];*FMR1* and *CYFIP1* mRNA expression levels: An amount of 2.5 mL of peripheral blood was collected in PAXgene RNA tubes, and total RNA was isolated using the PAXgene Blood RNA Kit (Qiagen, Valencia, CA, USA) according to the manufacturer’s instructions. RNA concentration was calculated using the Agilent 2100 Bioanalyzer system. cDNA synthesis and mRNA expression levels were carried out using 3 different concentrations (500, 250, 125 ng) in duplicate, as previously described [52]. Gene-specific *FMR1* or *CYFIP1* primers and probes and the reference genes, β-Glucuronidase (*GUS)* and Hypoxanthine-guanine phosphoribosyltransferase (*HPRT1)* were used and are as reported in [52];Plasma MMP-9 levels: MMP-9 levels (normalized with MMP-2 levels) were measured using the ELISA assay, MILLIPLEX MAP Human MMP Magnetic Bead Panel 2 (Merck Millipore, Billerica, MA, USA). The preparation of plasma samples and reagents was performed according to the manufacturer’s protocol. A total of 25 μL (1:20 dilution) of plasma samples were run in duplicates on Luminex^®^ plates, which included quality controls and negative and positive controls. The plates were run on Luminex^®^ with xPONENT 3.1 software, and the Median Fluorescent Intensity (MFI) data were analyzed using the spline curve-fitting method for calculating the concentrations of MMP-9 in each sample.

FMRP quantification: FMRP was quantified via the time-resolved fluorescence resonance energy transfer (TR-FRET) method using the Cisbio Human FMRP assay kit (Cisbio US, Bedford, MA, USA). Protease inhibitors were added to frozen peripheral blood mononuclear cells (PBMCs) during thawing. Then, cells were lysed in Cisbio lysis buffer supplemented with Benzonase (MilliporeSigma, Burlington, MA, USA) in the presence of MgCl_2_ to reduce viscoelasticity. Apart from these alterations, the manufacturer’s protocol was followed. After incubating fluorescent antibody conjugates with lysates at room temperature for 18 h, a control fibroblast fiducial line was used to fit a standard curve and interpolate percent change in fluorescence (ΔF%), as performed by Kim et al., 2019 [53]. A four-factor fit was used for ΔF% > 65, while a linear fit was used for ΔF% ≤ 65 to allow for interpolation of negative replicate values. Interpolated FMRP was then corrected to total protein loaded, as determined by BCA Protein Assay (Thermo Fisher Scientific, Rockford, IL, USA). Finally, relative FMRP was calculated by normalizing the mean of samples with control alleles.

### 2.3. Statistical Analysis

Statistical analyses of data were performed with open-source R software (version 4.2). Descriptive statistics were expressed as mean ± standard deviation (SD) of mean or median ± interquartile range (Q1 = 1st Quartile; Q3 = 3rd Quartile) for continuous variables and proportion (%) for categorical variables. Prior to statistical inferential tests, the Shapiro–Wilk test was used to check if a continuous variable follows a normal distribution. To measure the strength and direction of correlation between two variables, pairwise correlation analyses were performed by Spearman’s rank correlations because most variables were not normally distributed. Correlations were calculated using only complete data for both variables in pair and were adjusted for age as a covariate. Benjamini–Hochberg false discovery rate (FDR) method was applied to take multiple comparisons into account for a large number of pairwise correlations. However, for the nature of this exploratory analysis, given the small sample size relative to the number of all considered clinical and molecular measures, two-tailed *p*-values of less than 0.05 were considered statistically significant in this present study.

## 3. Results

The final study sample comprised 59 individuals (mean age 14.24 years, SD 5.92, range 6.00–31.69), with 55 of the male gender. Table 1 shows demographic, descriptive, clinical, and molecular biomarker measures for the study sample. The majority had a body mass index (BMI) in the normal range, with a mean of 22.87 (SD 6.56) and a median of 21.72 (Q1 = 17.12, Q3 = 28.28). The mean non-verbal IQ was 46.83 (SD 13.95), and the Adaptive Behavior Composite score on the VABS was 50.81 (SD 17.35).

Correlation analysis between molecular and clinical/Toolbox measures was performed, and the results are shown in Table 2, Figure 1 and Figure 2. There were positive correlations between MMP-9 levels with weight (r = 0.53, *p* = 0.0001) and BMI (r = 0.51, *p* = 0.0002). MMP-9 levels were also negatively correlated with the ADAMS manic/hyperactive behavior score (r = −0.31, *p* = 0.0347) (Appendix A).

There were also significant positive correlations between expression levels of *FMR1* mRNA and the VABS-3 Adaptive Behavior Composite (r = 0.43, *p =* 0.0034), the VABS-3 Communication score (r = 0.31, *p =* 0.0355), the VABS-3 Daily Living score (r = 0.43, *p =* 0.0028), and the VABS Socialization score (r = 0.41, *p =* 0.0047). Levels of *FMR1* mRNA also correlated positively with the Leiter Nonverbal IQ (r = 0.34, *p =* 0.022) as well as the ELS conversation sub-scale (r = 0.38, *p =* 0.0472). *FMR1* mRNA levels correlated positively with DCCS (r = 0.53, *p =* 0.0229) and Flanker (r = 0.56, *p =* 0.0045) from the Toolbox and negatively with the SNAP IV total score (r = −0.34, *p =* 0.0207) (Appendix A). 

A positive correlation was observed between *CYFIP1* mRNA levels and the ADAMS general anxiety (r = 0.36, *p* = 0.018) and with the ADOS-2 comparison score (r = 0.32, *p =* 0.0441) (Appendix A). Levels of FMRP did not correlate significantly with any of the clinical measures. 

Inter-correlation analysis among clinical measures (Figure 3) shows a positive correlation between non-verbal IQ and ELS (narration r = 0.64, *p* < 0.001; conversation r = 0.48, *p* = 0.003; and composite score r = 0.6, *p* < 0.001). Not surprisingly, the VABS adaptive composite correlated positively with non-verbal IQ (r = 0.65, *p* < 0.001) and the ELS (narration r = 0.57, *p* < 0.001; conversation r = 0.57, *p* < 0.001; and composite score r = 0.61, *p* < 0.001). The ADOS-2 comparison score was negatively correlated with CSHQ (r = −0.3, *p =* 0.031) (Figure 4).

As expected, there were strong correlations among several of the Toolbox measures, as seen in Figure 5. Ongoing and future studies focused on the Toolbox battery in FXS will determine whether the present composite scores are psychometrically supported for this specific population. 

## 4. Discussion

Although there have been several studies examining the cognitive and behavioral profile of FXS, in this study, we have investigated the interplay of molecular measures with the clinical phenotype observed in these patients. Our key results include the positive correlations between MMP-9 levels and weight and BMI and negative correlations with MMP-9 and hyperactive behavior. The ELS composite and conversation measures correlated with mRNA levels, and all of the ELS scores correlated with the Leiter IQ measure. The FXS mouse model is obese [54], and the majority of individuals with FXS overeat, which is, perhaps, related to stuffing their mouths, limited satiation, and/or obsessive thinking about food [55]. The most severe form of obesity in FXS is observed in those with the Prader–Willi-like phenotype (PWP), who present with lower IQ and a higher rate of autism compared to those with FXS without the PWP [34]. However, here we show for the first time a correlation between MMP-9 and weight in FXS. Levels of *FMR1* mRNA, which would be expected in those who are mosaic with some cells carrying unmethylated alleles, also correlated positively with markers of language, adaptive skills, Leiter IQ, and executive function. 

*CYFIP1* mRNA levels were associated with ADAMS General Anxiety and also with the ADOS-2 Comparison. Aberrant behavior, as measured by the ABC, correlated inversely with measures of cognition, including the Leiter, the VABS, and the ELS measures. This is consistent with prior studies showing that those with higher cognitive functioning have less severe aberrant behaviors [6,56,57]. Robust correlations between the Toolbox cognition battery and measures of adaptive behavior, non-verbal IQ, and language provide support for the clinical meaningfulness of this tool for research in FXS, adding to its applicability in IDD populations [36,42,43].

FMRP levels for the current group of subjects were all low, with a mean of approximately 20% of the mean among controls and some levels below the lower limit of detection. These low levels are not surprising, given that FMRP levels are generally quite low to undetectable in the full mutation range. However, it was important to determine whether any of the higher detected levels corresponded to higher-scoring clinical outcomes. The absence of significant correlations between FMRP and clinical measures could also reflect a lack of power in the analysis due to the low sample size, with only half of the samples (n = 32) being available for the determination of FMRP levels. It is important to note that this analysis is necessarily limited to males since FMRP levels are driven primarily by X-activation levels in females, where a fraction of cells display a normal molecular phenotype due to the expression of the normal *FMR1* allele. Importantly, FMRP values are low, with a very small range of values. The TR-FRET method determines FMRP levels by computing the difference in fluorescent signal between patient samples in buffer and buffer alone. Many individuals with FXS produce either no or very low levels of FMRP, with samples sometimes producing a fluorescence signal that is only 2–3% larger than that of the buffer background, thus preventing an accurate determination of FMRP level in those cases. Further confounding the detection issue is the fact that non-FMRP proteins may cross-react with the detection antibodies producing a small signal in the absence of FMRP. Therefore, detecting a true small increase from the background is challenging.

Our results showed good agreement between clinical objective and questionnaire-based measures, especially the ABC scale and its correlation with adaptive skills. The ELS measures each correlated significantly with the measures of nonverbal IQ and adaptive behaviors, replicating previous findings on different samples [35,39]. Such findings are further validations of the ELS measures as they reflect the integral bidirectional relationship of language and cognition over the course of development and the critical role of language in learning and performing a wide range of everyday tasks [58].

### Strengths and Limitations 

This paper has several weaknesses, including the low number of patients (n = 60) relative to a large number of clinical and molecular measures, although as data from the other sites of the trial are added, these numbers will increase in future reports. No significant trends were found after FDR adjustment for multiple comparisons. These data represent the baseline measures for a clinical trial of metformin in individuals with FXS, and future publications will have greater numbers and outcome measures pertinent to the trial results. The number of patients for which FMRP was available was limited, and most of the participants were males, so the range of FMRP measures was small, whereas the inclusion of more females would allow a greater expansion of the FMRP range.

## 5. Conclusions

In conclusion, our results identify relationships between clinical and molecular markers in individuals with FXS and important correlations among clinical measures providing further empirical support for the NIH Toolbox Cognition Battery and the ELS for this population. Of note, MMP-9 levels are associated positively with weight, while *FMR1* mRNA levels are associated with markers of better language, cognition, and adaptive skills. We also highlight the good correspondence between questionnaire-based measures to assess such behavior as the Aberrant Behavior Checklist and objective cognitive and adaptive behavior scales. Future studies can examine the implications of these molecular markers on the clinical phenotype of FXS and pave the way for targeted treatments of FXS.

## Figures and Tables

**Figure 1 cells-12-01920-f001:**
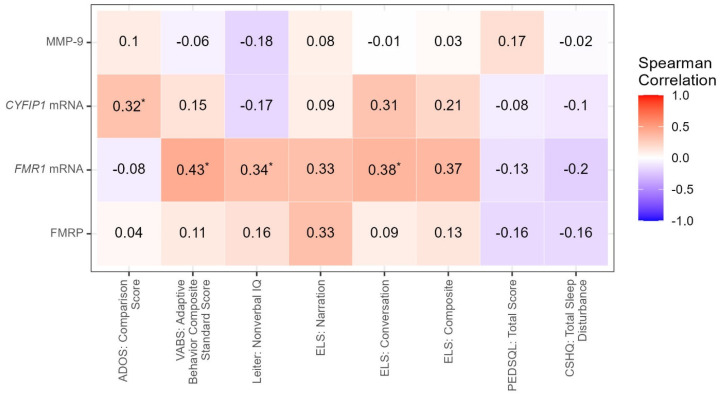
Correlations between Biomarkers and Clinical Measures. Molecular measures are indicated on the y-axis, and clinical measures are indicated on the x-axis. * Significant at <0.05; NS = not significant at *p*-value < 0.05.

**Figure 2 cells-12-01920-f002:**
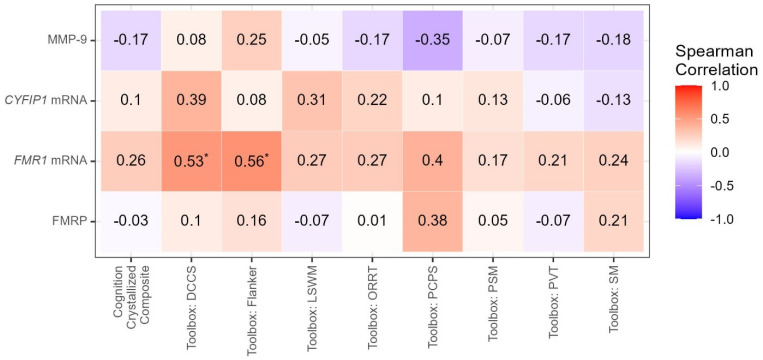
Correlations between Biomarkers and Toolbox Measures. Molecular measures are indicated on the y-axis, and clinical Toolbox measures are indicated on the x-axis. * Significant at <0.05; NS = not significant at *p*-value < 0.05.

**Figure 3 cells-12-01920-f003:**
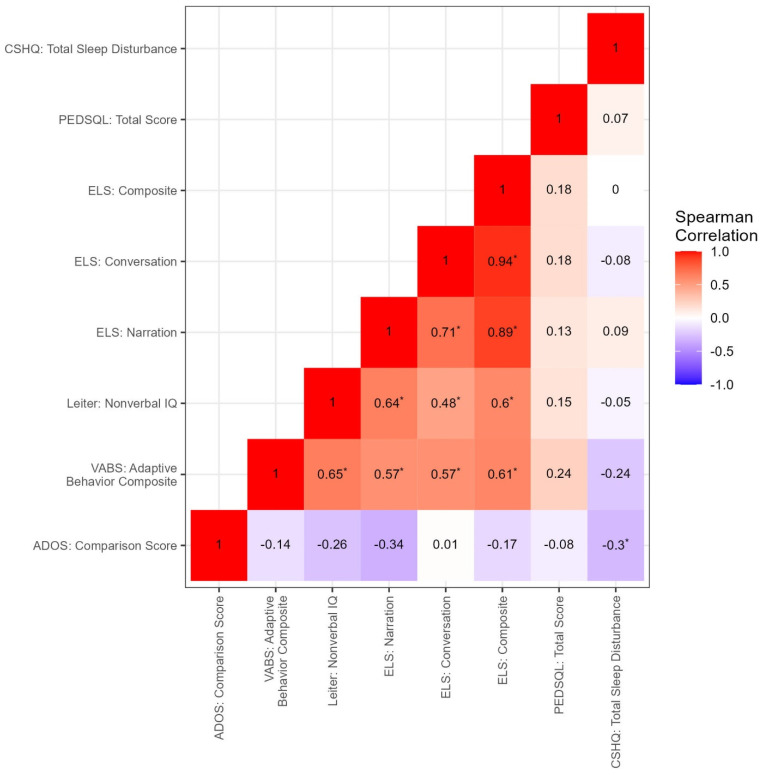
Correlations among Clinical Measures. Correlation coefficients depicting various clinical measures and their relationships. * Significant at *p*-value < 0.05.

**Figure 4 cells-12-01920-f004:**
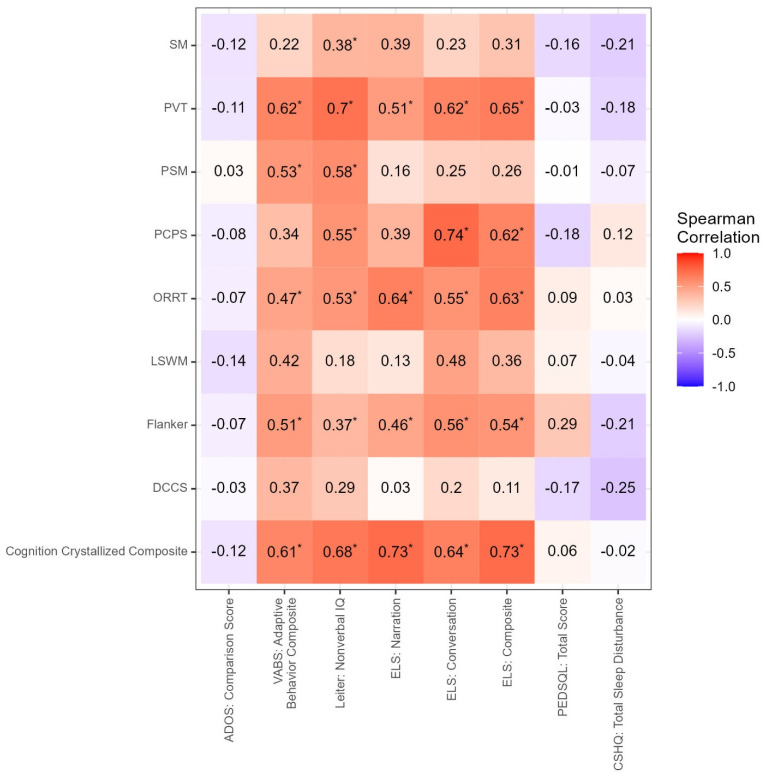
Correlations between Clinical and Toolbox Measures. Correlation between Toolbox measures and clinical measures with subtests of the Toolbox on the y-axis and clinical measures on the x-axis. * Significant at *p*-value < 0.05.

**Figure 5 cells-12-01920-f005:**
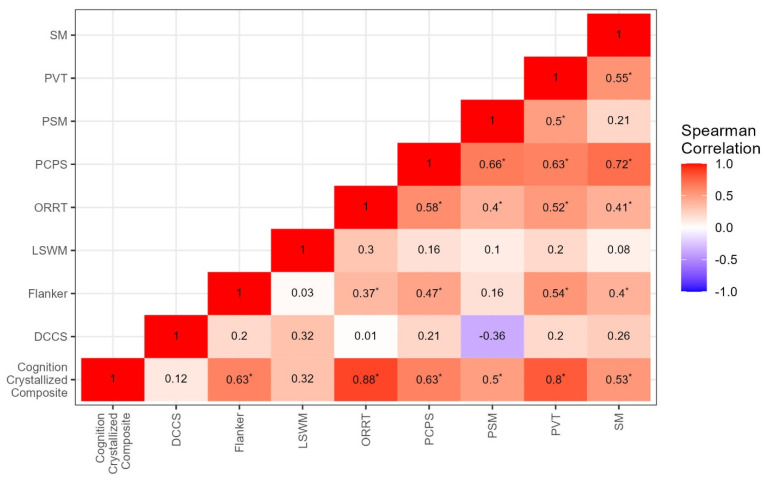
Correlations among Toolbox measures. Inter-correlations between subtests of the NIH Toolbox. * Significant at *p*-value < 0.05.

**Table 1 cells-12-01920-t001:** Summary of the Statistics of Clinical and Molecular Biomarker Measures.

Variable	N	Mean (SD) or N (%)	Median (Q1, Q3)
Age (years)	59	14.24 (5.92)	13.44 (9.25, 17.94)
Gender, Male	59	55 (93.2%)	
Weight (kg)	59	55.38 (26.30)	50.45 (32.62, 80.67)
BMI	59	22.87 (6.56)	21.72 (17.12, 28.28)
ABC			
Irritability	58	16.07 (13.88)	12 (6, 21)
Lethargy	58	7.38 (6.06)	5 (4, 10.75)
Stereotypy	58	6.55 (5.03)	6 (3, 9)
Hyperactivity	58	11.88 (7.47)	11 (5.25, 17)
Inappropriate Speech	58	5.78 (3.22)	6 (3, 9)
Social Avoidance	58	3.07 (3.05)	3 (0, 4)
ADAMS			
Manic/Hyperactive Behavior	55	7.56 (3.75)	8 (4.5, 10)
Depressed Mood	55	2.07 (2.46)	1 (0, 3)
Social Avoidance	55	7.89 (4.08)	8 (5, 11)
General Anxiety	55	6.85 (3.99)	7 (4, 10)
Obsessive Compulsive Behavior	55	2.73 (2.51)	2 (0.5, 4)
VABS			
Adaptive Behavior	57	50.81 (17.35)	54 (35, 63)
Communication	57	43.53 (19.33)	44 (24, 60)
Daily Living Skills	57	54.70 (23.61)	59 (34, 72)
Socialization	57	52.77 (18.61)	50 (38, 70)
Leiter: Nonverbal IQ	59	46.83 (13.95)	45 (34.5, 58)
SNAP-IV			
ADHD Combined Total	58	28.05 (12.25)	30 (18.25, 36)
ODD Total	58	5.59 (5.80)	4 (1.25, 8)
ELS			
Narration	36	57.83 (29.62)	52.5 (37.75, 73)
Conversation	36	75.47 (36.77)	74 (45, 97)
Composite	36	66.65 (30.71)	66 (42.75, 83.88)
ADOS-2: Comparison Score	53	7.32 (2.07)	8 (6, 9)
Toolbox			
Cognition Crystallized Composite	51	60.96 (13.12)	61 (54, 66.5)
DCCS	22	66.05 (19.17)	68 (55.5, 78.5)
Flanker	30	54.37 (26.97)	37 (31.25, 81.5)
LSWM	22	69.73 (13.63)	65 (59, 78)
ORRT	51	66.82 (13.32)	68 (57.5, 76)
PCPS	32	68.41 (21.45)	59.5 (54, 81.5)
PSM	32	82.53 (16.29)	79 (71, 91.25)
PVT	52	60.67 (15.05)	59.5 (51.75, 67)
SM	42	0.49 (0.23)	0.48 (0.33, 0.65)
Peds QL Total Score	58	65.87 (13.34)	67.11 (57.42, 74.61)
CSHQ: Total Sleep Disturbance score	58	46.40 (6.10)	47 (44, 50)
Molecular Category	53		
Full mutation		36 (67.9%)	
Meth mosaic		8 (15.1%)	
Size mosaic		9 (17.0%)	
MMP-9	52 (1)	0.53 (0.35)	0.41 (0.29, 0.64)
*CYFIP1* mRNA	48 (5)	0.36 (0.19)	0.33 (0.23, 0.42)
*FMR1* mRNA	48 (5)	0.30 (0.48)	0.07 (0, 0.55)
FMRP *	32	0.199 (0.255)	0.104 (0.062, 0.244)

Abbreviations: ABC: Aberrant Behavior Checklist (ABC); ADAMS: The Anxiety Depression and Mood Screen; VABS: Vineland Adaptive Behavioral Score; SNAP-IV: Swanson, Nolan, and Pelham Questionnaire; ADHD: Attention Deficit Hyperactivity Disorder; ODD: Oppositional Defiant Disorder; ELS: Expressive Language Sampling; ADOS-2: Autism Diagnostic Observation Scale 2; DCCS: Dimensional Change Card Sort Test; Flanker: Flanker Inhibitory Control and Attention Test; LSWM: List Sorting Working Memory Test; ORRT: Oral Reading Recognition Test; PCPS: Pattern Comparison Processing Speed Test; PSM: Picture Sequence Memory Test; PVT: Picture Vocabulary Test; SM: Speeded Matching; PedsQL: Pediatric Quality of Life Inventory; CSHQ: Children’s Sleep Habits Questionnaire. * FMRP values are normalized to the mean of samples with normal alleles to represent a relative ratio. For example, 0.21 = 21% FMRP compared to patients with control alleles.

**Table 2 cells-12-01920-t002:** Spearman’s Correlation Coefficients ^#^ between Clinical Measures and Biomarkers, adjusted for Age.

	MMP-9		*CYFIP1* mRNA		*FMR1* mRNA		FMRP	
Clinical Measure	r	*p*-Value	r	*p*-Value	r	*p*-Value	r	*p*-Value
Weight (kg)	0.53	<0.0001 *	0.28	NS	0.14	NS	0.04	NS
BMI	0.51	0.0002 *	0.22	NS	0.22	NS	0.05	NS
ABC								
Irritability	0.08	NS	−0.13	NS	−0.23	NS	0.03	NS
Lethargy	−0.14	NS	−0.05	NS	−0.23	NS	−0.19	NS
Stereotypy	0.08	NS	0	NS	−0.2	NS	0.21	NS
Hyperactivity	−0.01	NS	−0.04	NS	−0.26	NS	0.15	NS
Inappropriate Speech	0.04	NS	−0.06	NS	−0.27	NS	0.16	NS
Social Avoidance	0	NS	0.09	NS	−0.06	NS	−0.07	NS
ADAMS								
Manic/Hyperactive Behavior	−0.31	0.0347	0.16	NS	−0.23	NS	−0.15	NS
Depressed Mood	−0.11	NS	0.1	NS	−0.1	NS	0.08	NS
Social Avoidance	−0.28	NS	0.2	NS	−0.11	NS	−0.08	NS
General Anxiety	−0.03	NS	0.36	0.018	−0.11	NS	−0.09	NS
Obsessive Compulsive Behavior	−0.04	NS	0.13	NS	−0.02	NS	0.23	NS
VABS								
Adaptive Behavior Composite Standard Score	−0.06	NS	0.15	NS	0.43	0.0034 *	0.11	NS
Communication	−0.09	NS	0.22	NS	0.31	0.0355	0.06	NS
Daily Living Skills	−0.16	NS	0.02	NS	0.43	0.0028 *	0.09	NS
Socialization	0.17	NS	0.12	NS	0.41	0.0047 *	0.17	NS
Leiter: Nonverbal IQ	−0.18	NS	−0.17	NS	0.34	0.0226	0.16	NS
SNAP—IV: ADHD Combined Total	−0.26	NS	0.04	NS	−0.34	0.0207	−0.1	NS
SNAP—IV: ODD Total	−0.11	NS	−0.14	NS	−0.2	NS	−0.27	NS
ELS								
Narration	0.08	NS	0.09	NS	0.33	NS	0.33	NS
Conversation	−0.01	NS	0.31	NS	0.38	0.0472	0.09	NS
Composite	0.03	NS	0.21	NS	0.37	NS	0.13	NS
ADOS: Comparison Score	0.1	NS	0.32	0.0441	−0.08	NS	0.04	NS
Toolbox								
Cognition Crystallized Composite	−0.17	NS	0.1	NS	0.26	NS	−0.03	NS
DCCS	0.08	NS	0.39	NS	0.53	0.0229	0.1	NS
Flanker	0.25	NS	0.08	NS	0.56	0.0045 *	0.16	NS
LSWM	−0.05	NS	0.31	NS	0.27	NS	−0.07	NS
ORRT	−0.17	NS	0.22	NS	0.27	NS	0.01	NS
PCPS	−0.35	NS	0.1	NS	0.4	NS	0.38	NS
PSM	−0.07	NS	0.13	NS	0.17	NS	0.05	NS
PVT	−0.17	NS	−0.06	NS	0.21	NS	−0.07	NS
SM	−0.18	NS	−0.13	NS	0.24	NS	0.21	NS
PEDSQL: Total Score	0.17	NS	−0.08	NS	−0.13	NS	−0.16	NS
CSHQ: Total Sleep Disturbance score	−0.02	NS	−0.1	NS	−0.2	NS	−0.16	NS

Note: ^#^ Bivariate correlations on pairs of variables were estimated using pairwise complete observations. * Significant at FDR < 0.05; NS = not significant at *p*-value < 0.05. Abbreviations: see Table 1.

## Data Availability

Kyoungmi Kim and Randi Hagerman had full access to all the data in the study and took responsibility for the integrity of the data and the accuracy of the data analysis. Data can be shared on request and are also available at www.clinicaltrials.org (accessed on 9 August 2022).

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
