# Peer review of "Intercorrelation of Molecular Biomarkers and Clinical Phenotype Measures in Fragile X Syndrome"

_cells, 2023, doi:10.3390/cells12141920_

Round 1

Reviewer 1 Report

The manuscript presents the results of a statistical analysis conducted on a sample of 59 individuals with FXS (including 55 males), focusing on the correlation between molecular biomarkers, clinical measures, and cognitive performance. The study employed descriptive statistics and Spearman's rank correlations to analyze the data. The findings reveal significant (positive) correlations between MMP-9 levels and weight/BMI, as well as negative correlations between MMP-9 levels in plasma and ADAMS manic/hyperactive behaviour score. In addition, they show a statistically significant correlationbetween FMR1 mRNA levels in blood and several behaviour characteristics including adaptive behaviour, communication, daily ling skills, socialization and inhibitory control and attention deficits (an NIH iPAD-based cognition set of tests), as well as between CYFIP1 mRNA levels in blood and ASD characteristics (according to ADOS assessment test). Clearly, this is a first step towards the identification of valuable biomarkers for diagnosing the various behaviour phenotypes associated with FXS. However, the small sample size and the exploratory nature of the analysis warrant cautious interpretation. Further research with larger sample sizes is needed to validate and expand upon these initial findings.

Author Response

We thank the reviewer for his comments. The paper includes in the limitation and future study section, that future studies are warranted with larger sample sizes to validate, confirm, and expand upon these initial findings.

Reviewer 2 Report

The paper, “Intercorrelation of molecular biomarkers and clinical phenotype measures in Fragile X Syndrome” focuses on identifying potential clinical biomarkers of Fragile X Syndrome (FXS).  The authors hypothesized that the expression levels of MMP-9, FMR1, CYFIP1 mRNA, and FMRP protein may correlate with behavioral abnormalities in FXS.  The authors showed that several mRNA expressions correlated with the FXS phenotype.  However, I did come up with some concerns about the evidence they found in this manuscript. 

(1)

The authors showed the correlation r value and statistical p-value between mRNA expression and FXS phenotype.   However, no raw data (x-y scatter plots) are presented to explain this conclusion.  This makes it difficult to evaluate the conclusions. The authors should at least show plots for data that are significantly different. 

(2)

The authors showed a correlation between MMP-9 levels and body weight and BMI. This is interesting new evidence. However, it does not provide a hypothesis as to why MMP-9 is important in controlling body weight in FXS. The authors need to discuss the potential mechanisms of MMP-9 in weight control.

(3)

Although several mRNA expressions were shown to correlate with the FXS phenotype, the correlation of each mRNA expression should also be shown. It would also be good to show a three-dimensional correlation between MMP-9, FMR1, and FXS phenotypes. These results may provide stronger evidence that the expression patterns of MMP-9 and FMR1 are biomarkers of FXS severity. 

(4)

Another approach to understanding the characteristics between individuals is to use machine learning to perform a reduced dimensionality analysis (like principal component analysis). Have you considered using computational analysis to find biomarkers for FXS?

Author Response

REVIEWER 2

 The authors showed the correlation r value and statistical p-value between mRNA expression and FXS phenotype.   However, no raw data (x-y scatter plots) are presented to explain this conclusion.  This makes it difficult to evaluate the conclusions. The authors should at least show plots for data that are significantly different. 

We have generated the plots and are now shown in the Supplementary material

The authors showed a correlation between MMP-9 levels and body weight and BMI. This is interesting new evidence. However, it does not provide a hypothesis as to why MMP-9 is important in controlling body weight in FXS. The authors need to discuss the potential mechanisms of MMP-9 in weight control.

We have added a comment regarding the correlation between MMP-9 levels and body weight and BMI and also some references.

Although several mRNA expressions were shown to correlate with the FXS phenotype, the correlation of each mRNA expression should also be shown. It would also be good to show a three-dimensional correlation between MMP-9, FMR1, and FXS phenotypes. These results may provide stronger evidence that the expression patterns of MMP-9 and FMR1 are biomarkers of FXS severity. 

We have now included the plots showing the correlations between mRNAs (FMR1 and CYFIP1) and clinical measures and the correlations bewteen molecular markers.

Another approach to understanding the characteristics between individuals is to use machine learning to perform a reduced dimensionality analysis (like principal component analysis). Have you considered using computational analysis to find biomarkers for FXS?

Thanks for the question. The use of machine learning methods is to make dimensional reduction, group classifications, or predictions for high-dimensional data. However, this project focuses on only four biomarkers and their correlations with clinical measures. Therefore, we found that machine learning methods were not appropriate for the scope of this paper. We considered that correlation analysis was most appropriate for the intended objective of this paper (i.e., examine item-level correlations between biomarkers and clinical measures). Further, we believe that our case-only study design is not appropriate for biomarker discovery types of work without appropriate controls.

Reviewer 3 Report

The manuscript entitled "Intercorrelation of molecular biomarkers and clinical phenotype measures in Fragile X Syndrome," aims to determine the correlative relationship between four possible peripheral biomarkers with a range of NIH Toolbox measures that have been previously reported to be associated with Fragile X Syndrome patients. Indeed, the authors included a large array of clinical, cognitive, behavioural tests in the analysis and determined correlative values with each of the biomarkers (MMP-9, Fmr1 mRNA, CYFIP1 mRNA, and FMRP). Indeed, I agree with the authors that these types of studies are highly valuable in the field as the use of biomarkers early on can offer greater perspectives on the challenges patients can face either later in development or adulthood if not yet detected, and a greater potential of individualized treatment approaches. My primary concern with the manuscript was that, while the authors included an impressive, comprehensive battery of tests, some aspects of the methodological reporting was largely lacking making interpretation of the results hard to determine. Overall, the manuscript is interesting but requires more information than what is given. I have listed my more major concerns/queries, as well as minor points, below. 

Major Comments: 

1. Interpretation of the clinical measures reported in Table 1. Table 1 lists each of the variable tests in each category with the mean (with SD or percent of people included in the study) and the median of the results. While the authors included a brief description of each of the tests or test categories, they did not include information of the context of these numbers. Including what the score range of a neurotypic mean or median for each of the tests in the table would be highly advantageous to those not intimately familiar with each test and give the reader context for this data. 

2. Reporting of molecular measures. 

2A. For the Fmr1 and CYFIP1 mRNA analysis, the detail of the methodology used was largely inadequate. While it cited a study using similar protocols but basic information regarding the current study is also needed for proper interpretation. 

- how much blood was used from each participant for RNA isolation? RNA was isolated using a different kit than the cited paper, so did they still use the same amount for the PCR experiments or did they yield different quantities? 

- in the PCR experiments, what concentration of mRNA did they use to synthesize cDNA (there were 3 different concentrations in the paper cited)? Did they run technical replicates for this experiment? 

- the primer sequences of each should be included in the methods section. 

- Gold standard practices for quantitative QPCR is the use of more than one reference gene. Why was only one reference gene used and how did they determine this was an appropriate choice? 

2B. How the determination of MMP-9 in plasma was done was unclear.  The major concern here is that the authors chose to use a less quantitative method for determining protein amounts than more stringent methods widely used for determination of MMP-9 levels (ie. Western blotting, ELISA, etc). The reasoning for this is unclear as is much of the protocol used for the MFI results. The authors should address their reasoning for this in the manuscript as well as provide information regarding the questions below.  

- What was the starting amount of plasma, and the amount used for MFI analysis?

- Given that MFI is only relevant as a relative measurement, what were the baseline, positive or negative controls?  

- what was the antibody used in the detection and where was it acquired from? Did it differentiate between Pro-MMP9 or the activated form of the protein?

2C. Given that the participants were genetically screened for FXS, it is unclear what information the authors planned to gain from FMRP as a measure (unlike previous studies on pre-mutations, etc). As they reported within the manuscript, the values were predictably very low or non-detectable in all subjects, so it is unclear why this was included in this study. If the authors believe (as they have stated) that the number of participants for this part of the study is too low than why not wait until you have the number that you need as this is an ongoing study? The lack of any female data is also a major limitation to this. Both the increase in numbers and sex differences would add significant insight into this parameter.   

3. The authors included some interesting analysis looking at the possible correlations between different clinical measures presented in figure 3-5. However, assuming that all the measures were taken from the same set of individuals, this would have also been interesting to see with the biomarker measures. For example, how does differential MMP-9 values correlate to Fmr1 mRNA, etc. and how do differences across multiple biomarkers relate to clinical measures or the toolbox tests? This could potentially show added value in measuring multiple biomarkers. 

Minor Comments. 

1. pg 4, line 153, "tis" should be replaced with "this".

2. pg 4, line 156, "as" should be replaced with "and".

Author Response

REVIEWER 3

Major Comments: 

  1. Interpretation of the clinical measures reported in Table 1. Table 1 lists each of the variable tests in each category with the mean (with SD or percent of people included in the study) and the median of the results. While the authors included a brief description of each of the tests or test categories, they did not include information of the context of these numbers. Including what the score range of a neurotypic mean or median for each of the tests in the table would be highly advantageous to those not intimately familiar with each test and give the reader context for this data.

For the majority of the measures, there are no fragile X-specific standard score ranges, and due to the nature of the developmental disability, a comparison with the normal range may not be as helpful as just looking at the raw scores as individual baseline for a clinical trial. We added the general description of “higher scores mean higher impairment”, and the average range for the Leiter-3 nonverbal IQ.

For the Fmr1 and CYFIP1 mRNA analysis, the detail of the methodology used was largely inadequate. While it cited a study using similar protocols but basic information regarding the current study is also needed for proper interpretation. 

Additional information were added to the method section

- how much blood was used from each participant for RNA isolation? RNA was isolated using a different kit than the cited paper, so did they still use the same amount for the PCR experiments or did they yield different quantities? 

These details have been added to the text.

-in the PCR experiments, what concentration of mRNA did they use to synthesize cDNA (there were 3 different concentrations in the paper cited)? Did they run technical replicates for this experiment? 

We used 3 different conc. (500, 250ng, 125ng) and they were run in duplicates and this is reported the methods.

- the primer sequences of each should be included in the methods section.

The sequence of the primers and probe have been established, documented and used for over a decade. We have referenced it.

- Gold standard practices for quantitative QPCR is the use of more than one reference gene. Why was only one reference gene used and how did they determine this was an appropriate choice? 

Although we have run several studies and used one reference gene (GUS) that has shown to be pretty robust, we actually run 2 reference genes (and found no difference). We added this information to the method section.

How the determination of MMP-9 in plasma was done was unclear.  The major concern here is that the authors chose to use a less quantitative method for determining protein amounts than more stringent methods widely used for determination of MMP-9 levels (ie. Western blotting, ELISA, etc). The reasoning for this is unclear as is much of the protocol used for the MFI results. The authors should address their reasoning for this in the manuscript as well as provide information regarding the questions below.  

- What was the starting amount of plasma, and the amount used for MFI analysis? Given that MFI is only relevant as a relative measurement, what were the baseline, positive or negative controls?  

what was the antibody used in the detection and where was it acquired from? Did it differentiate between Pro-MMP9 or the activated form of the protein?

We added the information to the text. We used a standard Elisa kit- Millipore- (as specified in the method section), which contains all the reagents for the assay included the antibodies

2C. Given that the participants were genetically screened for FXS, it is unclear what information the authors planned to gain from FMRP as a measure (unlike previous studies on pre-mutations, etc). As they reported within the manuscript, the values were predictably very low or non-detectable in all subjects, so it is unclear why this was included in this study. If the authors believe (as they have stated) that the number of participants for this part of the study is too low than why not wait until you have the number that you need as this is an ongoing study? The lack of any female data is also a major limitation to this. Both the increase in numbers and sex differences would add significant insight into this parameter.   

We have added a paragraph on p15 that addresses these issues. Specifically, 

  • Despite the low FMRP levels in generalit was important to determine whether any of the higher detected levels corresponded to higher-scoring clinical outcomes. Although we agree with the reviewer that more samples would be better, because a substantial fraction of cases will always have undetectable FMRP values, due to the nature of full mutation alleles, it is not reasonable to expect that accessible numbers of patients would eliminate this situation. However, the FMRP levels constitute only one aspect of the study. 
  • (absence of females) It is important to note that this analysis is necessarily limitedto males, since FMRP levels are driven primarily on X-activation levels in females, where a fraction of cells display a normal molecular phenotype due to expression of the normal FMR1 allele - as serious confound for the study of FMRP expressed from full mutation alleles. 

 The authors included some interesting analysis looking at the possible correlations between different clinical measures presented in figure 3-5. However, assuming that all the measures were taken from the same set of individuals, this would have also been interesting to see with the biomarker measures. For example, how does differential MMP-9 values correlate to Fmr1 mRNA, etc. and how do differences across multiple biomarkers relate to clinical measures or the toolbox tests? This could potentially show added value in measuring multiple biomarkers. 

Minor Comments.  

  1. pg 4, line 153, "tis" should be replaced with "this".
  2. pg 4, line 156, "as" should be replaced with "and".

The appropriate corrections were made.

Reviewer 4 Report

The authors have submitted for review a paper entitled, “Intercorrelation of molecular biomarkers and clinical phenotype measures in Fragile X Syndrome.” The paper is well-written and covers an important topic. The paper is appropriate for publication in Cells after minor revisions.

Minor comments

1.     Line 335 – The citation for this statement appears to be incorrect.  Also, I am familiar with the fragile X mouse but have no knowledge of them being considered obese outside of special circumstances or age (e.g., doi.org/10.3390/cells11081350).  Please clarify and support such statement with a primary literature source.

2.     Most test descriptions in the Method section include the relationship between score and severity; however, this information is missing for a few, and the range of scores possible doesn’t seem to be listed. Because of this and because there are no control values, Table 1 mean scores are difficult to interpret. Also, aside from FMRP are the other molecular/protein measures in Table 1 normalized?  It only mentions it for FMRP.

3.     In Table 2, should the p-value for MMP-9 correlation with Manic/hyperactive behavior be given any asterisk?  On first read, I missed the significance of this correction, but noticed it in the Discussion.  Also, it seems that there could be a relationship between the correlations observed for MMP-9, hypoactivity, and weight/BMI, and this possibility could be mentioned.

4.     Is there anything to be discussed about the correlations involving Fmr1 mRNA with Flanker, etc. in the Discussion section?

5.     There are some odd statements and incomplete sentences in the descriptions of the different measures in the methods section.  For examples, see the ADOS-2 description.

6.     You may wish to specify in the abstract and otherwise at first mention which markers are measured as proteins (as opposed to mRNA, which is specified), to make this distinction clearer.

7.     Possibly re-visit where colons are placed in Figure legends.

Author Response

REVIEWER 4

 Line 335 – The citation for this statement appears to be incorrect.  Also, I am familiar with the fragile X mouse but have no knowledge of them being considered obese outside of special circumstances or age (e.g., doi.org/10.3390/cells11081350).  Please clarify and support such statement with a primary literature source.

A reference was provided.

  1. 2.     Most test descriptions in the Method section include the relationship between score and severity; however, this information is missing for a few, and the range of scores possible doesn’t seem to be listed. Because of this and because there are no control values, Table 1 mean scores are difficult to interpret.

Correlations between score and severity have now been included in the text

Also, aside from FMRP are the other molecular/protein measures in Table 1 normalized?  It only mentions it for FMRP.

FMRP was the only protein measured; however, RNA levels ( 2 genes)  and also MMP9 were normalized.

  1. In Table 2, should the p-value for MMP-9 correlation with Manic/hyperactive behavior be given any asterisk? On first read, I missed the significance of this correction, but noticed it in the Discussion. 

Correction was made

Also, it seems that there could be a relationship between the correlations observed for MMP-9, hypoactivity, and weight/BMI, and this possibility could be mentioned.

We added a comment in the Discussion

  1. Is there anything to be discussed about the correlations involving Fmr1 mRNA with Flanker, etc. in the Discussion section?

 We added this in Supplemental Figure 1.

  1. There are some odd statements and incomplete sentences in the descriptions of the different measures in the methods section. For examples, see the ADOS-2 description.

Correction was made. These have been corrected and expanded on where necessary, including the ADOS-2 description.

  1. You may wish to specify in the abstract and otherwise at first mention which markers are measured as proteins (as opposed to mRNA, which is specified), to make this distinction clearer.

We have made the distinction clearer in the text.

  1.     Possibly re-visit where colons are placed in Figure legends.

We have checked and we are not sure to which figure the reviewer is referring. The legends appear to be correct.

Round 2

Reviewer 2 Report

Thank you for the answers to the requested questions. 

I do not have further comments or requests. 

Reviewer 3 Report

Thank you for addressing my concerns. I have no further comments.